# Effective Perturbations on the Amplitude and Hysteresis of Erg-Mediated Potassium Current Caused by 1-Octylnonyl 8-[(2-hydroxyethyl)[6-oxo-6(undecyloxy)hexyl]amino]-octanoate (SM-102), a Cationic Lipid

**DOI:** 10.3390/biomedicines9101367

**Published:** 2021-10-01

**Authors:** Hsin-Yen Cho, Tzu-Hsien Chuang, Sheng-Nan Wu

**Affiliations:** 1Department of Physiology, National Cheng Kung University Medical College, Tainan 70101, Taiwan; lilyzhou861126@gmail.com (H.-Y.C.); fytg55qq@gmail.com (T.-H.C.); 2Institute of Basic Medical Sciences, National Cheng Kung University Medical College, Tainan 70101, Taiwan

**Keywords:** SM-102 (1-octylnonyl 8-[(2-hydroxyethyl)[6-oxo-6-(undecyloxy)hexyl]amino]-octanoate), erg-mediated K^+^ current, inwardly rectifying K^+^ current, current kinetics, voltage-dependent hysteresis, pituitary cell, Leydig cell, microglial cell

## Abstract

SM-102 (1-octylnonyl 8-[(2-hydroxyethyl)[6-oxo-6-(undecyloxy)hexyl]amino]-octanoate) is an amino cationic lipid that has been tailored for the formation of lipid nanoparticles and it is one of the essential ingredients present in the Moderna^TM^ COVID-19 vaccine. However, to what extent it may modify varying types of plasmalemmal ionic currents remains largely uncertain. In this study, we investigate the effects of SM-102 on ionic currents either in two types of endocrine cells (e.g., rat pituitary tumor (GH_3_) cells and mouse Leydig tumor (MA-10) cells) or in microglial (BV2) cells. Hyperpolarization-activated K^+^ currents in these cells bathed in high-K^+^, Ca^2+^-free extracellular solution were examined to assess the effects of SM-102 on the amplitude and hysteresis of the erg-mediated K^+^ current (I_K(erg)_). The SM-102 addition was effective at blocking I_K(erg)_ in a concentration-dependent fashion with a half-maximal concentration (IC_50_) of 108 μM, a value which is similar to the K_D_ value (i.e., 134 μM) required for its accentuation of deactivation time constant of the current. The hysteretic strength of I_K(erg)_ in response to the long-lasting isosceles-triangular ramp pulse was effectively decreased in the presence of SM-102. Cell exposure to TurboFectin^TM^ 8.0 (0.1%, *v*/*v*), a transfection reagent, was able to inhibit hyperpolarization-activated I_K(erg)_ effectively with an increase in the deactivation time course of the current. Additionally, in GH_3_ cells dialyzed with spermine (30 μM), the I_K(erg)_ amplitude progressively decreased; moreover, a further bath application of SM-102 (100 μM) or TurboFectin (0.1%) diminished the current magnitude further. In MA-10 Leydig cells, the I_K(erg)_ was also blocked by the presence of SM-102 or TurboFectin. The IC_50_ value for SM-102-induced inhibition of I_K(erg)_ in MA-10 cells was 98 μM. In BV2 microglial cells, the amplitude of the inwardly rectifying K^+^ current was inhibited by SM-102. Taken together, the presence of SM-102 concentration-dependently inhibited I_K(erg)_ in endocrine cells (e.g., GH_3_ or MA-10 cells), and such action may contribute to their functional activities, assuming that similar in vivo findings exist.

## 1. Introduction

SM-102 (heptadecan-9-yl 8-((2-hydroxyethyl)(6-oxo-6-(undecyloxy)hexyl)amino)octanoate, 4-hydroxybutyl)azanediyl)bis(hexane-6,1-diyl)bis(2-hexyldecanoate, 1-octylnonyl 8-[(2-hydroxyethyl)[6-oxo-6-(undecyloxy)hexyl]amino]-octanoate) is a synthetic and ionizable amino lipid that has been widely used in combination with other lipids in the formation of lipid nanoparticles [1,2]. The administration of luciferase mRNA in SM-102-containing lipid nanoparticles was previously reported to induce hepatic luciferase expression in mice [3]. Formulations containing SM-102 have been noticeably used in the development of lipid nanoparticles for the delivery of mRNA-based vaccines [2,4,5], since such an efficient transfection procedure, based on compacted lipopolyamine-coated plasmids, has been developed [6]. Moreover, SM-102 is known to be one of the ingredients in the Moderna^TM^ COVID-19 vaccine (https://www.fda.gov/media/144638/download) (accessed on 27 September 2021) [7]. However, it has been recently demonstrated that myocarditis might emerge following COVID-19 vaccination [8,9,10,11,12]. SM-102, one of the ingredients in Moderna^TM^’s vaccine, has been linked to the occurrence of myocarditis following COVID-19 vaccination [11,12,13,14]. However, the need of whether SM-102 exerts any perturbations on the magnitude of transmembrane ionic currents still remains unmet.

The erg-mediated K^+^ current (I_K(erg)_) is recognized to be encoded by three different subfamilies of the *KCNH* gene, which can give rise to the pore-forming α-subunit of erg-mediated K^+^ (i.e., K_erg_ or K_V_11) channels [15,16]. This type of ionic currents is regarded as constituting the cloned counterpart of the rapidly activating delayed-rectifying K^+^ currents enriched in heart cells, and the *KCNH2* gene encodes the pore-forming α-subunit of the voltage-gated K^+^ channels, K_V_11.1, commonly referred to as hERG [15,16]. The I_K(erg)_ is inherent in neurons or different types of electrically excitable cells, such as endocrine and neuroendocrine cells, and it can potentially affect the maintenance of the resting potential and the modifications in the subthreshold excitability [15,17,18,19,20]. Previous work has also noticeably demonstrated the ability of I_K(erg)_’s magnitude to regulate tumor cell apoptosis and proliferation [21]. However, whether or how the presence of SM-102 or other structurally similar compounds produces any modifications on the magnitude in this type of K^+^ currents is largely uncertain.

Previous reports have shown that pituitary cells (e.g., GH_3_ cells), in addition to the presence of voltage-gated K^+^ currents, exhibit I_K(erg)_ with an inwardly rectifying property. This type of ionic current was characterized by voltage-dependent activation and was proposed to be an important determinant of the resting membrane potential and, therefore, of cell excitability [17,18,22]. The resultant perturbations in this K_V_ channel activity also have the propensity to modify the stimulus-secretion coupling in these cells [18].

In view of the foregoing considerations, the following attempts were undertaken to evaluate how SM-102 or other relevant compounds (e.g., TurboFectin^TM^) could result in any modifications of ionic currents (e.g., erg-mediated K^+^ current (I_K[erg]_)) in pituitary tumor (GH_3_) cells. Spermine is known to be a polyamine and polycationic compound which can suppress the inwardly rectifying properties of the inward rectifier K^+^ current (I_K(IR)_), while TurboFectin^TM^ is a proprietary mixture of a broad-spectrum protein/polyamine with histones and lipids which is known to be a transfection reagent. Similar to SM-102, they are polyamine groups with cationic properties. Moreover, previous studies have demonstrated a pertinent link of I_K(erg)_ to cellular apoptosis or proliferation. We decided to investigate the possible action of SM-102 on another type of endocrine cells (i.e., MA-10 Leydig cells). Findings from the present observations enable us to point out that I_K(erg)_ inherent in different cell types could be an important target through which SM-102 or TurboFectin^TM^ can act to influence the functional activities of the cells involved.

## 2. Materials and Methods

### 2.1. Drugs, Chemicals and Solutions Used in This Study

SM-102 (9-Heptadecanyl 8-((2-hydroxyethyl)(6-oxo-6-(undecyloxy)hexyl)amino)octanoate, 4-hydroxybutyl)azanediyl)bis(hexane-6,1-diyl)bis(2-hexyldecanoate), 1-octylnonyl 8-[(2-hydroxyethyl)[6-oxo-6-(undecyloxy)hexyl]amino]-octanoate, C_44_H_87_NO_5_, https://www.caymanchem.com/product/33474/sm-102 (accessed on 27 September 2021), CAS registry number: 2089251-47-6) was acquired from Cayman (Asia Bioscience, Taipei, Taiwan), while TurboFectin^TM^ 8.0 transfection reagent (Cat number: TF81001, https://www.origene.com/products/others/transfection-reagents/turbofectin) (accessed on 27 September 2021) was from OriGene (Level Biotechnology, Tainan, Taiwan). TurboFectin^TM^ is a proprietary mixture of a broad-spectrum protein/polyamine with histones and lipids, and it was stored at 4 °C to avoid the development of precipitates. Spermine (*N*,*N*′-bis(3-aminopropyl)-1,4-diaminobutane) and tetrodotoxin (TTX) were acquired from Sigma-Aldrich (Merck, Taipei, Taiwan), whereas E-4031 (N-[4-[[1-[2-(6-methyl-2-pyridinyl)ethyl]-4-piperidinyl]carbonyl]phenyl]methanesulfonamide dihydrochloride) and PD118057 (2-[[4-[2-(3,4-dichlorophenyl)ethyl]phenyl]amino]benzoic acid) were from Tocris (Union Biomed, Taipei, Taiwan). Glyceryl nonivamide (GLNVA) was synthesized as described previously [23], and it was kindly provided by Dr. Yi-Ching Lo (Department of Pharmacology, Kaohsiung Medical University, Kaohsiung, Taiwan). The chemical structures of SM-102, spermine and GLNVA are shown in the Appendix A. To make stock solution keep at −20 °C, SM-102 was dissolved in chloroform, spermine in water and GLNVA in dimethyl sulfoxide (DMSO), while TurboFectin^TM^ was in 80% ethanol solution. During the experiments, chloroform, DMSO or ethanol in the desired concentration of each compound was diluted with bathing or pipette solution to be less than 0.01%. Chlorotoxin was provided by Dr. Woei-Jer Chuang (Department of Biochemistry, National Cheng Kung University Medical College, Tainan, Taiwan). All other chemicals were commercially available and of reagent grade. Double-distilled water was deionized using a Milli-Q water purification system (Merck).

The composition of normal Tyrode’s solution was as follows (in mM): NaCl 136.5, KCl 5.4, CaCl_2_ 1.8, MgCl_2_ 0.53, glucose 5.5 and HEPES-NaOH 5 (pH 7.4). To measure I_K(erg)_, high-K^+^, Ca^2+^-free solution contained (in mM): KCl 130, NaCl 10, MgCl_2_ 3, glucose 6 and HEPES-KOH (pH 7.4). To record inwardly rectifying K^+^ currents (I_K(IR)_), the pipettes were filled up with solution (in mM): K-aspartate 140, KH_2_PO_4_ 1, MgCl_2_ 1, EGTA 0.1, Na_2_ATP 3, Na_2_GTP 0.1 and HEPES-KOH 5 (pH 7.2). In some experiments, the recording pipette was filled with a solution containing SM-102 (100 μM), TurboFectin^TM^ (0.1% *v*/*v*) or spermine (30 μM).

### 2.2. Cell Preparations

GH_3_ pituitary tumor cells, acquired from the Bioresources Collection and Research Center ((BCRC-60015); Hsinchu, Taiwan), were maintained in Ham’s F-12 medium supplemented with 15% horse serum, 2.5% fetal calf serum and 2 mM L-glutamine; the MA-10 cell line, which was originally derived from a mouse Leydig tumor, was grown in the Waymouth medium (HyClone^TM^) containing 10% fetal bovine serum (FBS) [24,25], while BV2 microglial cells were maintained in Dulbecco’s modified Eagle medium supplemented with 10% FBS [26]. Cells (e.g., GH_3_, MA-10 or BV2 cells) were grown in a humidified environment of 5% CO_2_/95% air. A colorimetric method was commonly used in examining cell densities in microtiter plates with a tetrazolium salt (WST) and an ELISA reader (DYNATECH, Chantilly, VA, USA). The experiments were performed 5 or 6 days after cells were subcultured (60–80% confluence).

### 2.3. Electrophysiological Measurements

On the day of the measurements, we dispersed cells with a 1% trypsin-EDTA solution, and a few drops of cell suspension were rapidly placed in a custom-made chamber firmly affixed on the stage of an inverted DM-II microscope (Leica; Major Instruments, Kaohsiung, Taiwan). Cells were allowed to be immersed at room temperature (20–25 °C) in normal Tyrode’s solution, the composition of which was detailed above, and they were allowed to settle on the chamber’s bottom before the measurements were performed. The patch pipettes were prepared from Kimax-51 borosilicate glass tubing (#34500; Kimble, Dogger, New Taipei City, Taiwan) using a vertical electrode puller (PP-830; NARISHIGE, Major Instruments, Tainan, Taiwan). The recording electrodes used had a tip diameter of 1 μM and a resistance of 3–5 MΩ. During the recordings, they were mounted in an air-tight holder, which had a suction port on the side, and a chloride silver wire was used to make contact with the internal electrode solution. We recorded varying types of ionic currents in the whole-cell mode of a modified patch-clamp technique with dynamic adaptive suction (i.e., a decremental change in suction negative pressure in response to progressive increase in the seal resistance) with the aid of either an Axoclamp-2B (Molecular Devices, Sunnyvale, CA, USA) or an RK-400 amplifier (BioLogic, Claix, France), as described elsewhere [18,27]. This procedure was allowed to increase the success rate of gigaseal formation as well as to reduce loss of the gigaseal in long-term recordings. For whole-cell current recordings, following seal formation (>1 GΩ), cell membrane was, thereafter, ruptured by gentle suction. The liquid junction potentials, that arose when the composition of the pipette solution differed from that in the bath, were zeroed shortly before seal formation was created, and the whole-cell data were corrected.

### 2.4. Whole-Cell Current Recordings

The signals were monitored and stored online at 10 kHz in an ASUS ExpertBook laptop computer (P2451F; ASUS, Tainan, Taiwan) equipped with a Digidata 1440A interface (Molecular Devices). During the measurements with analog-to-digital and digital-to-analog conversion, the latter device was operated using pCLAMP 10.6 software (Molecular Devices) ran under Microsoft Windows 7 (Redmond, WA, USA). The laptop computer was placed on the top of an adjustable Cookskin stand (Ningbo, Zhejiang, China) to allow efficient manipulation during the operation. Through digital-to-analog conversion, the voltage-clamp protocols with varying rectangular commands or voltage ramp waveforms (i.e., piecewise linear voltage versus time) were specifically designed and suited for determining the steady-state or instantaneous relationship of current versus voltage (*I–V*) and voltage-dependent hysteresis of the current (e.g., I_K(erg)_) in GH_3_ and MA-10 cells.

### 2.5. Data Analyses

To calculate percentage inhibition of SM-102 on I_K(erg)_ in GH_3_ or MA-10 cells, the cells examined were stepped from −10 to −100 mV with a duration of 1 s, the peak amplitude of deactivating I_K(erg)_ during application of SM-102 was compared with the control value (i.e., SM-102 was not present). The concentration-dependent relation of SM-102 on the inhibition of I_K(erg)_ amplitude was properly fitted with the modified Hill equation. That is,
(1)$$\mathrm{Percentage}\;\mathrm{inhibition}\;\left(\%\right)=\frac{E_{\max}\times {\left[\mathrm{SM}\right]}^{n_{H}}}{IC_{50}^{n_{H}}+{\left[\mathrm{SM}\right]}^{n_{H}}}$$
where [SM] represents the concentration of SM-102 applied; n_H_ or IC_50_ is the Hill coefficient or the concentration required for a 50% inhibition of I_K(erg)_, respectively; E_max_ is SM-102-induced maximal inhibited of I_K(erg)_ amplitude. This equation reliably converged to produce the best-fit line and parameter estimates in Figure 2D.

The time-dependent rate constant of blocking (k_+1_*) or unblocking (k_−1_) was determined from the deactivation time constants (τ_deact_) of I_K(erg)_ activated by the long hyperpolarizing pulses from −10 to −90 mV. The τ_deact_ value achieved in the control period (i.e., SM-102 was not present) or during cell exposure to different concentrations of SM-102 was determined by fitting the trajectory of each current trace with a single exponential function. Because a Hill coefficient of about 1 was obtained according to the concentration–response curve, the blocking or unblocking rate constant was further estimated using the relation:(2)$$\frac{1}{\tau_{\mathrm{deact}}}=k_{+1}^{*}\times \left[\mathrm{SM}\right]+k_{-1}$$
where [SM] is the SM-102 concentration used, and k_+1_* or k_−1_, respectively, result from the slope and the y-axis intercept at [SM] = 0 of the liner regression in situations where the reciprocal time constants of I_K(erg)_ deactivation (i.e., 1/τ_deact_) versus different SM-102 concentrations were derived.

### 2.6. Curve-Fitting Procedures and Statistical Analyses

Curve fitting (linear or non-linear (e.g., exponential or sigmoidal fitting)) to experimental data sets was carried out with the goodness of fit by using various maneuvers, such as Microsoft “Solver” add-in function embedded in Excel^TM^ 2019 (Microsoft^®^) ran under Office 365^®^ (Microsoft^®^) and 64-bit OriginPro^®^ 2016 program (OriginLab; Scientific Formosa, Kaohsiung, Taiwan). The data are presented as the mean ± standard error of the mean (SEM), with sample sizes (n) indicating the number of GH_3_ or HL-1 cells from which the data were collected. The Student’s t-test (paired or unpaired) and one-way analysis of variance (ANOVA) were utilized for statistical analyses; however, as the difference among different groups was necessarily evaluated, post hoc Duncan’s multiple-range comparisons were further exploited. Moreover, assuming that normality underlying ANOVA was violated, a non-parametric Kruskal–Wallis test was used. Differences between the values were considered significant when *p* < 0.05.

## 3. Results

### 3.1. Effect of SM-102 on Erg-Mediated K^+^ Current (I_K(erg)_) Measured from GH_3_ Cells

In the first stage of the whole-cell current recordings, we measured the possible effects of SM-102 on the amplitude of I_K(erg)_ activated by long-lasting step hyperpolarization with a duration of 1 s. During the measurements, we kept cells immersed in high-K^+^, Ca^2+^-free solution in which 1 μM tetrodotoxin (TTX) and 0.5 mM CdCl_2_ were contained, and the recording electrode was filled up with a K^+^-containing solution. The compositions in these solutions were stated in Materials and Methods. As demonstrated in Figure 2A, a family of large inward currents on the 1 s step hyperpolarization from −10 to −90 mV could be robustly detected in these cells. Such a hyperpolarizing voltage pulse was noticed to elicit an instantaneous current followed by a voltage- and time-dependent decay of the K^+^ inward current, as reported elsewhere [17,20,28]. This type of ionic current, which was effectively inhibited by either E-4031 or glyceryl nonivamide (GLNVA), has been identified to be an erg-mediated K^+^ current (I_K(erg)_) [17,18,20,29]. For example, 1 min after GH_3_ were exposed to 10 μM E-4031, the peak or sustained I_K(erg)_, respectively, decreased to 42 ± 6 or 21 ± 8 pA (*n* = 7) from control values of 177 ± 42 or 49 ± 9 pA (*n* = 7). Meanwhile, the presence of 10 μM GLNVA resulted in a reduction in the peak or sustained I_K(erg)_ from 173 ± 39 or 47 ± 8 pA (*n* = 7) to 48 ± 7 or 23 ± 8 pA (*n* = 7), respectively. Moreover, 1 min after GH_3_ cells were exposed to SM-102 at a concentration of 100 or 300 μM, the peak (i.e., initial) or sustained (i.e., end-pulse) component of deactivating I_K(erg)_ evoked by a long hyperpolarizing pulse from −10 to −90 mV was progressively depressed. Figure 1 illustrates the time course of the SM-102-induced inhibition of peak I_K(erg)_. For example, as the rectangular voltage step from −10 to −90 mV with a duration of 1 s was delivered to the examined cell (indicated in the inset of Figure 2A) to activate I_K(erg)_, the application of 300 μM SM-102 was noticed to result in a conceivable reduction in the peak or sustained amplitude of I_K(erg)_ to 119 ± 34 or 22 ± 4 pA (*n* = 8, *p* < 0.05) from control values of 174 ± 43 or 46 ± 7 pA (*n* = 8), respectively. After a washout of SM-102, the initial I_K(erg)_ was reversed to 169 ± 39 pA (*n* = 7). Furthermore, as cells were exposed E-4031 (10 μM) alone, the peak amplitude of I_K(erg)_ evoked by the same voltage-clamp protocol was almost abolished, as evidenced by a reduction in the current amplitude to 19 ± 4 pA (*n* = 7, *p* < 0.01) from a control value of 171 ± 41 pA (*n* = 7). Similar inhibitory effects on I_K(erg)_ in GH_3_ cells were obtained as cells were exposed to 10 μM GLNVA alone.

### 3.2. Concentration-Dependent Analysis of SM-102-Mediated Inhibition of I_K(erg)_ in GH_3_ Cells

The biophysical properties of I_K(erg)_ elicited by step hyperpolarization in the presence of different SM-102 concentrations tended to display a concentration-dependent increase in the deactivation rate of the current (Figure 2A,B). In other words, under our experimental conditions, as such a population of erg-mediated K^+^ (K_erg_) channels was perturbed by the presence of SM-102, the current flowing through those channels would change (or relax) over time to a new equilibrium level at a rate which may reflect the underlying kinetics of the channels. For these reasons, we further evaluated the detailed kinetics of the SM-102-induced inhibition of I_K(erg)_ recorded from GH_3_ cells. The concentration-dependence of the decay in deactivating I_K(erg)_ caused by SM-102 is illustrated in Figure 2A,B. Although the initial component of deactivating I_K(erg)_ elicited by a hyperpolarizing pulse from −10 to −90 mV mildly decreased during cell exposure to SM-102, its inhibitory effects on I_K(erg)_ were noticed to be a concentration-dependent increase in the rate of current deactivation together with a reduction in the residual, steady-state current. For example, as cells were 1 s hyperpolarized from −10 to −90 mV, the deactivation time constant (τ_deact_) of I_K(erg)_ in the presence of 100 or 300 μM SM-102 was well approximated by a single-exponential with the value of 84.9 ± 9.4 or 45.9 ± 8.7 ms (*n* = 9), respectively (Figure 2C). As such, increasing SM-102 concentration noticeably, not only reduced the peak component of I_K(erg)_, but it also had the propensity to enhance the apparent deactivation rate of the current. The results enabled us to reflect that the inhibitory effect of SM-102 on I_K(erg)_ in GH_3_ cells could be reasonably accounted for by a state-dependent block in situations where it had an interaction particularly at the open state of the channel. Moreover, according to the reaction binding scheme, the blocking or unblocking rate constant of current decay in the presence of SM-102 was calculated to be 0.0503 s^−1^μM^−1^ or 6.753 s^−1^, respectively; the K_D_ (k_−1_/k_+1_*) for the SM-102-mediated increase in the deactivation rate of the current was well approximated to be 134.3 μM, a value that was nearly identical with its IC_50_ value, i.e., 108 μM (Figure 2D). These results, therefore, strengthened the notion that a concentration-dependent increase by SM-102 in the rate of I_K(erg)_ deactivation in response to a rectangular hyperpolarizing pulse could largely account for the reduction in I_K(erg)_ amplitude detected in GH_3_ cells.

### 3.3. Inhibitory Effect of SM-102 on Current versus Voltage (I–V) Relationship of I_K(erg)_

We continued to investigate the effect of SM-102 on I_K(erg)_ measured at the different levels of membrane potentials. As demonstrated in Figure 3, in the control period (i.e., SM-102 was not present), the currents decayed at a voltage more negative than −60 mV, and current relaxation was noticed to become faster with a greater step hyperpolarization, as reported previously [17,18,20]. Cell exposure to SM-102 at a concentration of 300 μM progressively decreased the amplitude of I_K(erg)_ in response to membrane hyperpolarizations (Figure 3B,C). The averaged *I–V* relationship of I_K(erg)_ taken at the beginning or end of hyperpolarizing pulses is illustrated in Figure 3B or Figure 3C, respectively. The current amplitudes in Figure 3B between the absence and presence of 300 μM SM-102 were significantly different at voltages more negative than −80 mV, while those in Figure 3C significantly differed at the voltages more negative than −50 mV. It is important to note here that in addition to a reduction in the initial component of I_K(erg)_, the *I–V* relationship of sustained I_K(erg)_ (Figure 3C) was shifted to less negative potentials during exposure to SM-102 (300 μM).

### 3.4. Effect of SM-102 on Voltage-Dependent Hysteresis of I_K(erg)_

The voltage-dependent hysteresis of I_K(erg)_ has been previously characterized in GH_3_ cells [28,29], and the hysteretic strength of different types of ionic currents has been recently disclosed to exercise an important impact on electrical behaviors of electrically excitable cells [29,30,31]. Analyses of hysteretic behavior have proven to be a useful tool in studies of ion channels [31,32]. Therefore, we next continued to investigate whether or how cell exposure to SM-102 could perturb the hysteretic behavior in response to the isosceles-triangular ramp pulse in GH_3_ cells. In these whole-cell recording experiments, the examined cell was voltage-clamped at −10 mV and a set of the upright isosceles-triangular ramp pulses in the range of −110 to 0 mV with a total ramp duration of 3.2 s (or ramp speed of ±0.069 mV/ms) was designed and then applied to it through digital-to-analog conversion (Figure 4A). In particular, current amplitude elicited by the upsloping (ascending) and downsloping (descending) limbs of the triangular ramp pulse was actually distinguishable. The observations indicated a voltage-dependent hysteresis for the elicitation of I_K(erg)_, as demonstrated in Figure 4A, on the basis of the relationship of instantaneous I_K(erg)_ versus membrane voltage. Additionally, by adding 100 μM SM-102, I_K(erg)_ activated by the upsloping limb of the long-lasting triangular ramp decreased, to a lesser extent, than that measured from the downsloping limb. For example, in the control period (i.e., SM-102 was not present), I_K(erg)_ at the level of −80 mV elicited upon the upsloping and downsloping ends of triangular ramp pulse were −71 ± 12 and −396 ± 34 pA (*n* = 8), respectively, the values of which were found to differ significantly between them (*p* < 0.05). Moreover, in the presence of 100 μM SM-102, the amplitude of ascending and descending I_K(erg)_ at the isopotential level (−80 mV) evidently decreased to −58 ± 10 and −285 ± 31 pA (*n* = 8, *p* < 0.05), respectively. Therefore, the magnitude of SM-102-induced I_K(erg)_ inhibition at the upsloping and downsloping limb of such a triangular ramp also differed significantly (*p* < 0.05). The application of 100 μM SM-102 to cells led to a reduction in I_K(erg)_ amplitude (measured at −80 mV) during the upsloping or downsloping limb of the triangular ramp pulse by about 18% or 28%, respectively.

As denoted by the dashed arrows in Figure 4, from a difference in the area (i.e., Δarea) encircled by the hysteretic curves elicited at the ascending (upsloping) and descending (downsloping) direction, we further quantified the degree of the voltage-dependent hysteresis of I_K(erg)_ responding to a 3.2 s triangular ramp pulse. The hysteretic area of I_K(erg)_ occurring in GH_3_ cells conceivably decreased in the presence of SM-102. Figure 4B summarizes the data demonstrating effects of SM-102 (100 and 300 μM), TurboFectin^TM^ (i.e., TurboFectin^TM^ 8.0), SM-102 plus PD118057 and TurboFectin^TM^ plus PD118057 on the area taken between these two curves. For example, in addition to its decrease in I_K(erg)_ amplitude, the presence of 100 or 300 μM SM-102 decreased the area responding to the long-lasting triangular ramp, as evidenced by a respective reduction in Δarea to 22.2 ± 4.5 mV nA (*n* = 8, *p* < 0.05) or 16.3 ± 4.1 mV nA (*n* = 8, *p* < 0.05) from a control value of 26.1 ± 5.1 mV nA. Meanwhile, the addition of TurboFectin^TM^ (0.3%) was able to decrease the value of Δarea from 26.1 ± 5.1 to 16.5 ± 3.9 mV nA (*n* = 8, *p* < 0.05). In the continued presence of 300 μM SM-102 or 0.3% TurboFectin^TM^, a further application of PD118057 (10 μM) was capable of reversing the hysteretic area decreased by exposure to SM-102 or TurboFectin^TM^. PD118057 was previously reported to be an activator of I_K(erg)_ [33], while TurboFectin^TM^ is a transfection agent which contains polyamine with histones and lipids (https://www.origene.com/products/others/transfection-reagents/turbofectin, assessed on 26 August 2021). As such, the presence of SM-102 could decrease I_K(erg)_ amplitude in a hysteresis-dependent fashion.

### 3.5. Effect of Intracellular Dialysis with SM-102 or Spermine on the Amplitude of I_K(erg)_

Spermine is known to be a polyamine compound and, under certain pathologic conditions, its intracellular concentration could vary with the metabolic status [34]. As it was enriched in a cell interior, spermine or spermidine has been disclosed to be an endogenous inhibitor of the inwardly rectifying K^+^ current (I_K(IR)_) or Ca^2+^-activated K^+^ currents [26,34,35]; we, then, intended to explore whether in GH_3_ cells dialyzed with SM-102 or spermine, I_K(erg)_ could be sensitive to being perturbed. In these experiments, when the whole-cell mode was substantially established, the examined cells were voltage-clamped at −10 mV, and we applied a downsloping ramp pulse to them. For the initial stage of measurements, we filled up the recording pipet with 140 mM CsCl, as membrane rupture occurred, the consecutive ramp pulse from −10 to −90 mV with a duration of 1 s was applied at 0.01 Hz and, under our experimental conditions, the I_K(erg)_ amplitude at the 20th pulse was almost abolished. Furthermore, as demonstrated in Figure 5A, as cells were dialyzed with 100 μM SM-102, I_K(erg),_ in response to such a downsloping ramp pulse, progressively decreased when the repetitive ramp pulse was applied at a rate of 0.01 Hz. Shortly after the whole-cell mode was established, the current amplitude at the 1st pulse measured at the level of −80 mV was 313 ± 24 pA (*n* = 8), a value which was significantly different from that at the 20th pulse (i.e., 3 min after drastic membrane rupture) (189 ± 19 pA; *n* = 8, *p* < 0.05). Meanwhile, the inwardly rectifying property of the current, which appeared at the voltage ranging between −100 and −70 mV, became progressively evident. However, in control cells (i.e., cells dialyzed without SM-102), the I_K(erg)_ amplitude measured at −80 mV between the 1st and 20th pulses was not noticed to differ significantly (309 ± 23 pA (1st pulse) versus 311 ± 22 pA (20th pulse); *n* = 8, *p* > 0.05). Additionally, in cells dialyzed with TurboFectin^TM^ (0.1%) or spermine (30 μM), the amplitude of I_K(erg)_ evoked by such a downsloping ramp pulse gradually diminished (Figure 5B,C(a–c)). The results of our experiment showed that, as GH_3_ cells were continually dialyzed with either SM-102, TurboFectin^TM^ or spermine, the I_K(erg)_ amplitude could be reduced, in combination with the increased inwardly rectifying property of the current.

### 3.6. Effect of SM-102 on I_K(erg)_ Identified in MA-10 Leydig Cells

It was previously demonstrated that some metal cations can influence the testosterone production on in vitro Leydig cells [36]. The magnitude of the voltage-gated K^+^ current (e.g., I_K(erg)_) has also been disclosed to modify tumor cell apoptosis and proliferation [21,37]. For these reasons, we next investigated whether I_K(erg)_ could be functionally expressed in another type of endocrine cells (i.e., MA-10 cells) and how it was modified by the presence of SM-102. In the measurements similar to those conducted above in GH_3_ cells, we kept MA-10 cells bathed in high-K^+^, Ca^2+^-free solution containing 1 μM TTX and 0.5 mM CdCl_2_, and the recording electrode was filled up with a K^+^-enriched solution. As demonstrated in Figure 6, the I_K(erg)_ present in MA-10 cells measured at various voltage steps substantially decreased in the presence of 300 μM SM-102. For example, as cells were continually exposed to 300 μM SM-102 and the voltage step from −10 to −110 mV was applied, the I_K(erg)_ taken at the beginning or end pulse of the rectangular hyperpolarizing pulse significantly decreased to 535 ± 69 pA (*n* = 8, *p* < 0.05) or 120 ± 24 pA (*n* = 8, *p* < 0.05) from the control values of 832 ± 87 pA (*n* = 8) or 309 ± 44 pA (*n* = 8), respectively. The averaged *I–V* relationships for a transient or late component of deactivating I_K(erg)_ taken with and without the addition of SM-102 were, respectively, demonstrated in Figure 6B,C. Current amplitudes in Figure 6B,C between the absence and presence of 300 μM SM-102 significantly differed at the voltages more negative than −60 mV. Moreover, during cell exposure to 300 μM SM-102, the τ_deact_ value of I_K(erg)_ activated from −10 to −110 mV decreased to 106 ± 17 ms from a control value of 312 ± 22 ms. (*n* = 8, *p* < 0.05). The IC_50_ value required for the SM-102-mediated inhibition of I_K(erg)_ observed in MA-10 cells was estimated to be 98 μM (Figure 6D). It was reasonable to assume, therefore, that similar to the experimental observations found in GH_3_ cells, the I_K(erg)_ inherently in MA-10 cells was actually inhibited by the presence of SM-102.

### 3.7. Effect of SM-102 on Inwardly Rectifying K^+^ Current (I_K(IR)_) in BV2 Microglial Cells

Intracellular dialysis with spermine was noticed to inhibit the magnitude of I_K(IR)_ in microglial cells [26]. In a final set of experiments, we further wanted to test whether SM-102 could produce any effects on I_K(IR)_ present in microglial cells [26,38]. As demonstrated in Figure 7A,B, the application of SM-102 drastically reduced the amplitude of I_K(IR)_ evoked in response to the 1 s upsloping ramp pulse from −120 to +100 mV. For example, cell exposure to SM-102 at a concentration of 100 or 300 μM significantly decreased the I_K(IR)_ amplitude measured at +50 mV to 132 ± 18 (*n* = 7, *p* < 0.05) or 99 ± 15 pA (*n* = 7, *p* < 0.05) from a control value of 161 ± 23 pA (*n* = 7). However, in contrast, a bath application of chlorotoxin (1 μM), an inhibitor of Cl^−^ channels, was not noticed to exercise any effect on I_K(IR)_ in these cells, although the I_K(IR)_ amplitude effectively diminished in cells dialyzed with spermine (30 μM) (Figure 7B).

## 4. Discussion

In the present study, cell exposure to SM-102 was able to decrease the amplitude of I_K(erg)_ and raise the deactivate rate of the current in GH_3_ and MA-10 cells. A concentration-dependent inhibition of I_K(erg)_ with an effective IC_50_ value of 108 μM was observed in the presence of different SM-102 concentrations. The K_D_ value derived from a quantitative description of the deactivating time course of I_K(erg)_ was well approximated to be 134 μM, indicating that these two values were quite similar. The exposure to this compound not only decreased the I_K(erg)_ magnitude, but also shifted the *I–V* relationship of the sustained I_K(erg)_ to less negative potentials. Furthermore, its presence effectively decreased the magnitude of the voltage-dependent hysteretic strength of I_K(erg)_ activated by an isosceles-triangular ramp pulse. In GH_3_ cells dialyzed with SM-102, TurboFectin^TM^ or spermine, the magnitude of I_K(erg)_ progressively diminished. The I_K(erg)_ or I_K(IR)_ identified, respectively, in either MA-10 Leydig cells or BV2 microglial cells were actually subject to be inhibited by the presence of SM-102. Collectively, findings from this study reflected that SM-102 or TurboFectin^TM^ tended to not be inert, and that their effectiveness in the perturbations of ionic currents demonstrated here could be a notable confounding factor which could perturb the functional activities of excitable cells occurring in vitro or in vivo.

During GH_3_-cell exposure to varying SM-102 concentrations, the deactivation rate of I_K(erg)_ activated by a long hyperpolarizing command voltage was evidently enhanced. As such, the SM-102 or TurboFectin^TM^ molecule tended to accelerate I_K(erg)_ deactivation in a time-, concentration- and state-dependent manner, suggesting that they reached the blocking site of the channel, particularly as the channel resides in a high-conducting open state (conformation). Indeed, according to the minimal binding scheme, the K_D_ value required for the SM-102-mediated acceleration of I_K(erg)_ deactivation rate was estimated to be 134 μM.

The intriguing phenomenon of voltage-dependent hysteresis of ionic currents including I_K(erg)_ has been proposed to play roles in influencing cell behaviors of varying types of electrically excitable cells [28,29,30,31,32]. In keeping with previous observations [28,29], the I_K(erg)_ present in GH_3_ or MA-10 cells did undergo voltage-dependent hysteresis, as it was activated by a long isosceles-triangular ramp voltage. Such hysteresis may result from a mode shift, because the voltage sensitivity of the gating charge movement is thought to rely on the previous state of the channel owing to the so-called dynamic voltage dependence. Furthermore, the instantaneous and non-equilibrium property inherent in I_K(erg)_ elicited by the isosceles-triangular ramp voltage could be, conceivably, modified in the presence of SM-102 or TurboFectin^TM^. Our results also found the effectiveness of SM-102 or TurboFectin^TM^ in diminishing the Δarea of the hysteretic loop for I_K(erg)_ in response to a long triangular ramp pulse. Therefore, under this scenario, any modifications of I_K(erg)_ caused by the presence of SM-102 or TurboFectin^TM^ were dependent not simply on the concentration of SM-102 or TurboFectin^TM^ applied, but also on varying confounding factors that included the pre-existing resting potential or different firing patterns of action potentials existing in non-voltage-clamped cells.

Because polyamines (e.g., spermine) have been previously reported to exist in cells at submillimolar concentrations, the putative activation gating behavior is assumed to result from their slow blocking and unblocking of the inwardly rectifying K^+^ (Kir) channel [34]. Thus, on depolarization, what was previously called “deactivation” corresponded to polyamines, causing a time-dependent decrease in the outward current. On hyperpolarization, the inward Kir current first increased time-independently due to fast Mg^2+^ unblocking and, then, increased in a time-dependent fashion, which was referred to as “activation”, was due to polyamine unblocking [26,34]. Alternatively, in our study, as GH_3_ cells were dialyzed with spermine (a polyamine compound), the magnitude of I_K(erg)_ gradually reduced in combination with an increase in the inwardly rectifying property. Furthermore, intracellular dialysis with SM-102 or TurboFectin^TM^ presented herein was able to produce a progressive decrease in I_K(erg)_, leading us to imply that the presence of polyamine ingredients would be highly engaged in this perturbation (i.e., rectification properties) occurring through a similar internal blocking mechanism.

The concentration-dependent relationship of the SM-102 effect on the I_K(erg)_ amplitude found in MA-10 Leydig cells was also performed with an effective IC_50_ value of 98 μM. We did test different compounds (e.g., E-4031 and GLNVA) for the inhibition of I_K(erg)_ in GH_3_ and MA-10 cells [17,18,20]. However, it would be interesting to perform gene-knock down experiments for testing if SM-102 could specifically block I_K(erg)_, although by using such a maneuver, other types of K^+^ channels have the propensity to potentially be modified. Since the experimental conditions determined in the present study were acute in the onset and reversible, other types of K^+^ currents which might influence the experimental results could be relatively negligible. However, the detailed mechanism of SM-102 actions on I_K(erg)_ as well as on the apoptosis and/or proliferation assay still needs to be further delineated.

BV2 cells have been demonstrated to functionally express classical Kir channels (i.e., Kir2.1 channels) [26]. This type of K^+^ channel is also enriched in heart cells [34,39]. It is worth noting that, since the magnitude of both I_K(IR)_ and I_K(erg)_ was widely expressed in heart cells [15,16,33,39], the inhibitory effectiveness of SM-102 or TurboFectin^TM^ in altering I_K(IR)_ and/or I_K(erg)_ may potentially participate in the functional activities of heart function. These polycationic molecules (e.g., spermine and spermidine) enter the K_ir_ or K_erg_ channel pore from the intracellular side and block K^+^ ion movement through the channel at depolarized potentials, thereby ensuring the longer plateau phase of the cardiac action potential [39]. However, to what extent SM-102-mediated perturbations on membrane ionic currents confer their effectiveness in the adverse effects of mRNA-based vaccines (e.g., Moderna^TM^) has yet to be further delineated. Whether the working concentrations of SM-102 or TurboFectin^TM^ used for their direct adjustments on ionic currents could be achieved in vitro or in vivo also remains to be determined.

## Figures and Tables

**Figure 1 biomedicines-09-01367-f001:**
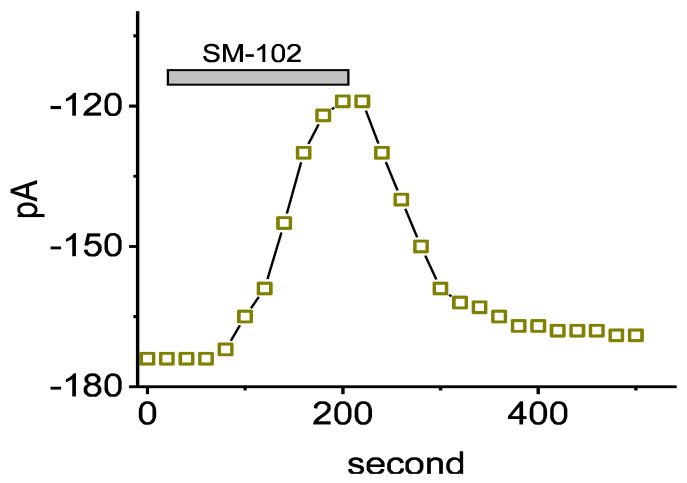
Time course showing effect of SM-102 (300 μM) on I_K(erg)_ recorded from GH_3_ cells. Cells were bathed in high-K^+^, Ca^2+^-free solution and the recording pipette was filled in a K^+^-enriched (145 mM) solution. Each current (indicated in open squares) was evoked by the hyperpolarizing pulse from −10 to −90 mV with a duration of 1 s at a rate of 0.05 Hz, and the amplitude at the beginning of hyperpolarizing pulse was measured. The horizontal bar shown above indicates the application of 300 μM SM-102.

**Figure 2 biomedicines-09-01367-f002:**
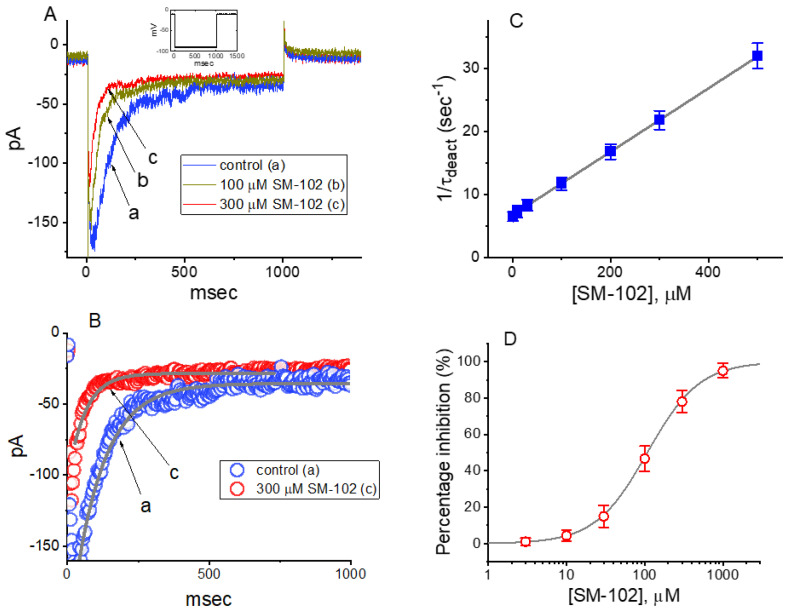
Effect of SM-102 on erg-mediated K^+^ current (I_K(erg)_) measured from pituitary tumor (GH_3_) cells. This series of experiments were conducted in cells which were bathed in high-K^+^, Ca^2+^-free solution containing 1 μM TTX and 0.5 mM CdCl_2_, and we filled up the electrode by using K^+^-containing (145 mM) solution. (**A**) Representative I_K(erg)_ traces obtained in the control period (i.e., SM-102 were not present; a) during cell exposure to 100 μM SM-102 (b) or 300 μM SM-102 (c). The voltage-clamp protocol applied is shown in inset. (**B**) Deactivation time courses of I_K(erg)_ taken in the absence (a) and presence of 300 μM SM-102 (c). The current trajectory (i.e., traces a and c) taken from Figure 2A was well fitted by a single exponential (gray line). Data points (indicated in open blue or red circles) with or without the SM-102 addition were reduced by 20 for clear illustration. (**C**) Kinetic evaluation of SM-102-mediated inhibition of I_K(erg)_ measured from GH_3_ cells (mean ± SEM; *n* = 7–8 for each point). The reciprocal of deactivation time constant of I_K(erg)_ (1/τ_deact_) taken on the basis of exponential fit of the I_K(erg)_ trajectory was properly derived and plotted against the SM-102 concentration in a linear manner (gray line). Blocking (k_+1_*) or unblocking (k_−1_) rate constant for binding scheme, given by either the slope or the y-axis intercept of the interpolated line, was 0.0503 s^−1^μM^−1^ or 6.753 s^−1^, respectively, and the K_D_ value (k_−1_/k_+1_* = 134.3 μM) was, thereafter, yielded. (**D**) Concentration-dependent relationship of SM-102 on I_K(erg)_ evoked by 1 s long membrane hyperpolarization (mean ± SEM; *n* = 8 for each point). Current amplitude (i.e., the peak amplitude of deactivating current) was measured at the start of each hyperpolarizing step from −10 to −90 mV with a duration of 1 s. The sigmoidal curve denotes the goodness-of-fit to the Hill equation, as described in Materials and Methods.

**Figure 3 biomedicines-09-01367-f003:**
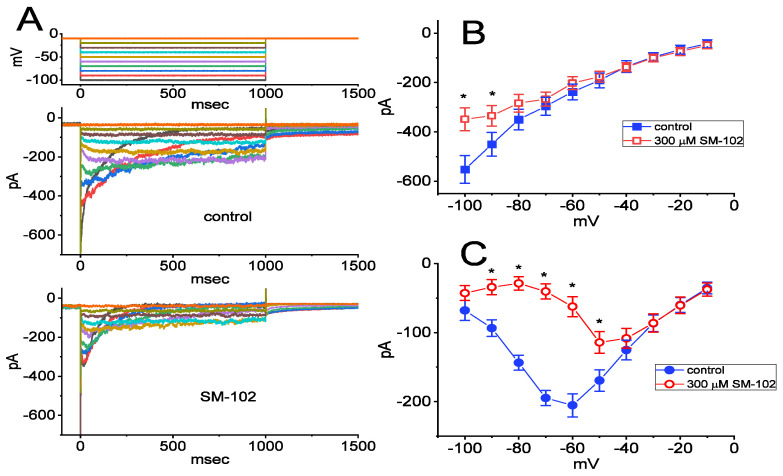
Inhibitory effect of SM-102 on current versus voltage (*I–V*) relationship of I_K(erg)_ identified in GH_3_ cells. In these experiments, cells were allowed to be bathed in high-K^+^, Ca^2+^-free solution containing 1 μM TTX and 0.5 mM CdCl_2_ and we filled up the electrode by using K^+^-enriched solution. (**A**) Representative current traces activated in response to various steps (indicated in the uppermost part) from a holding potential of −10 mV. The current traces in the upper part are controls (i.e., SM-102 was not present), while those in the lower part were obtained as the cell was exposed to 300 μM SM-102. (**B**,**C**) indicate the averaged *I–V* relationships of I_K(erg)_ taken at the beginning and end point of each hyperpolarizing pulse, respectively. ■: control; ○: in the presence of 300 μM SM-102. Each point represents the mean ± SEM (*n* = 9–10). * shown in (**B**,**C**) indicate the significant difference from control currents (*p* < 0.05) measured at the same level of the voltage. The statistical analyses were undertaken by ANOVA-2 for repeated measures, P (factor 1, groups among data taken at different level of membrane potentials) < 0.05, *p* (factor 2, groups between the absence and presence of SM-102) < 0.05, *p* (interaction) < 0.05, followed by post hoc Duncan’s multiple-range test, *p* < 0.05).

**Figure 4 biomedicines-09-01367-f004:**
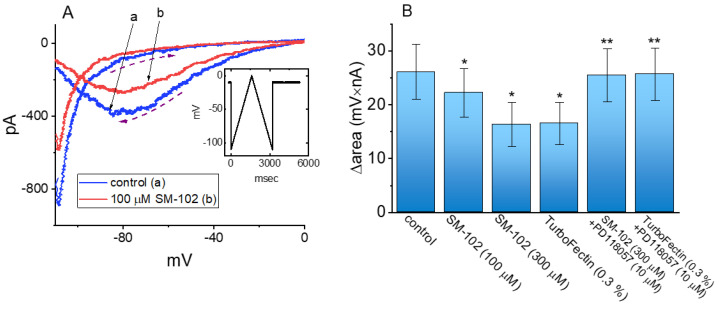
Effect of SM-102 on voltage-dependent hysteresis of I_K(erg)_ identified in GH_3_ cells. In this set of experiments, as whole-cell configuration was firmly established (i.e., membrane rupture after suction), we applied an isosceles-triangular ramp pulse to the examined cells. Inset indicates the voltage-clamp protocol used. (**A**) Representative instantaneous *I–V* relationship of I_K(erg)_ evoked by such triangular ramp pulse. a: control (i.e., SM-102 was not present); b: in the presence of 100 μM SM-102. The dashed arrows indicate the current trajectory which time passed over. (**B**) Summary bar graph showing effects of SM-102, TurboFectin^TM^, SM-102 plus PD118057 and TurboFectin^TM^ plus PD118057 on the hysteretic area (i.e., Δarea) of I_K(erg)_ elicited in response to long isosceles triangular ramp pulse (mean ± SEM; *n* = 8 for each bar). Data analysis was performed by ANOVA-1 (*p* < 0.05). * Significantly different from control (*p* < 0.05) and ** significantly different from SM-102 (300 μM) alone group (*p* < 0.05).

**Figure 5 biomedicines-09-01367-f005:**
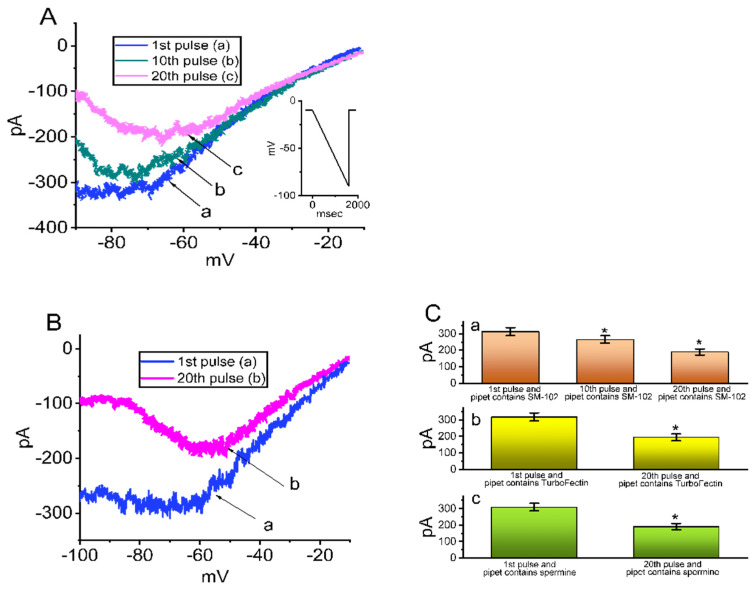
Effect of intracellular dialysis with SM-102, TurboFectin^TM^ or spermine on I_K(erg)_ in GH_3_ cells. In this set of measurements, we filled the recording electrode with an internal solution which contained 100 μM SM-102, 0.1% TurboFectin^TM^ or 30 μM spermine. In (**A**), as whole-cell mode was established, we applied a 2 s downsloping ramp pulse from −10 to −90 mV at a rate of 0.1 Hz. Inset at the right corner denotes the voltage-clamp protocol applied. In cells dialyzed with 100 μM SM-102, current trace labeled a is the one when the first pulse was applied (i.e., immediately after membrane rupture occurred), while that labeled b or c was obtained at the 10th or 20th pulse, respectively. Of note, there was a progressive decrease in I_K(erg)_ amplitude taken particularly at the voltages between −90 and −50 mV. In (**B**), cells were dialyzed with 0.1% TurboFectin^TM^, and the current trace labeled a or b activated by the same ramp pulse in (**A**) was taken at the 1st or 20th pulse, respectively. (**C**) Summary bar graph showing effect of intracellular dialysis with 100 μM SM-102 (a), 0.1% TurboFectin^TM^ (b) or 30 μM spermine (c) on I_K(erg)_ amplitude (mean ± SEM; *n* = 7 for the vertical bar in each panel). Data analysis in (**C**a) was performed by ANOVA-1 (*p* < 0.05), while those in (**C**b) and (**C**c) were by paired *t*-test (*p* < 0.05). *Significantly different from control (i.e., the amplitude taken at the first pulse).

**Figure 6 biomedicines-09-01367-f006:**
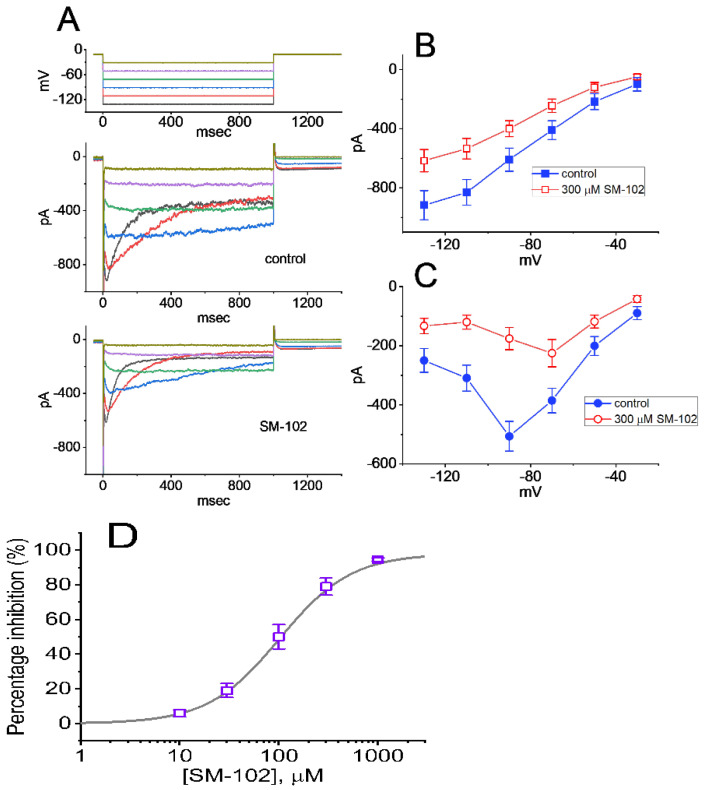
Effect of SM-102 on I_K(erg)_ recording from MA-10 Leydig cells. In these experiments, MA-10 cells were dispersed and then bathed in high-K^+^, Ca^2+^-free solution, and the recording electrode used was filled up with K^+^-enriched solution. (**A**) Representative I_K(erg)_ traces obtained in the absence (upper) and presence (lower) of 300 μM SM-102. In (**B**) or (**C**), the *I–V* relationship of peak or late (or sustained) I_K(erg)_ was demonstrated, respectively. Current amplitude was measured at the start (**B**) or end (**C**) of each hyperpolarizing pulse. Closed (square or circle) symbols in (**B**,**C**) are controls (i.e., SM-102 was not present), while open (square or circle) symbols were taken during cell exposure to 300 μM SM-102. Each point represents the mean ± SEM (*n* = 8). (**D**) Concentration-dependent inhibition of SM-102 on the amplitude of I_K(erg)_ taken from MA-10 cells. Current amplitude was measured at the start of hyperpolarizing pulse from −10 to −90 mV (mean ± SEM; *n* = 8 for each point).

**Figure 7 biomedicines-09-01367-f007:**
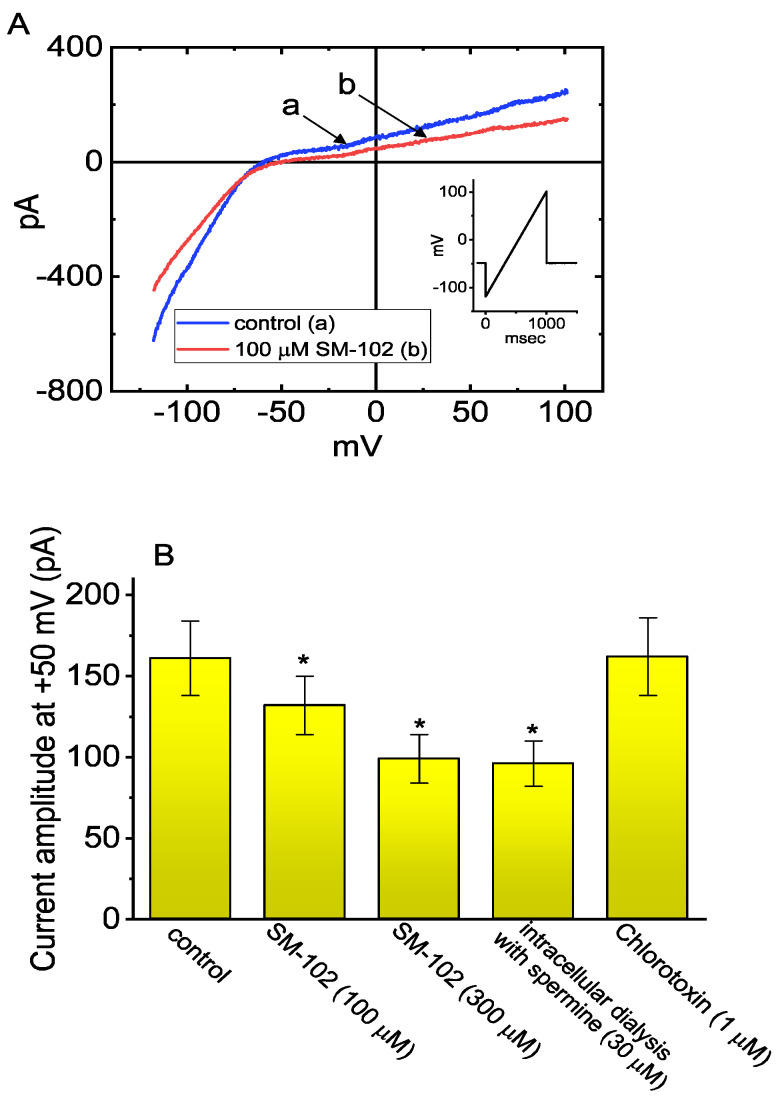
Effect of SM-102 on the inwardly rectifying K^+^ current (I_K(IR)_) in BV2 microglial cells. In this set of experiments, a holding potential was maintained at −50 mV and an upsloping ramp pulse from −120 to +100 mV with a duration of 1 s was applied to the cell examined. (**A**) Representative current traces obtained in the absence (a) or presence (b) of 100 μM SM-102. Inset shows the voltage protocol used. (**B**) Summary bar graph showing effect of SM-102, intracellular dialysis with spermine, or chlorotoxin on I_K(IR)_ amplitude activated by ramp pulse (mean ± SEM; *n* = 7 for each bar). Each current amplitude was measured at the level of +50 mV. Data analysis was performed by ANOVA-1 (*p* < 0.05). * Significantly different from control (*p* < 0.05).

## Data Availability

The original data are available upon reasonable request to the corresponding author.

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
