# Peer review of "Effective Perturbations on the Amplitude and Hysteresis of Erg-Mediated Potassium Current Caused by 1-Octylnonyl 8-[(2-hydroxyethyl)[6-oxo-6(undecyloxy)hexyl]amino]-octanoate (SM-102), a Cationic Lipid"

_biomedicines, 2021, doi:10.3390/biomedicines9101367_

Round 1
Reviewer 1 Report
General comments
Cho and colleagues present a paper entitled “Effective perturbations on the amplitude, gating and hysteresis of erg-mediated potassium current caused by 1-octylnonyl 8-[(2-hydroxyethyl)[6-oxo-6-(undecyloxy)hexyl]amino]-octanoate (SM-102), a cationic lipid”.
The authors have tested an ingredient present in the ModernaTM Covid-19 vaccine (SM-102), probably linked to the occurrence of myocarditis following the Covid-19 vaccination. In particular, they have evaluated SM-102 effects in GH3 (a cell line derived from a rat pituitary tumor), MA-10 (cell line derived from a mouse Leydig tumor) and microglia cells; all cells are not linked with myocarditis, anyway.
However, the manuscript is well written and a detailed electrophysiological analysis was conducted.
The authors demonstrated that SM-102 inhibits erg-mediated K+ current (IK(erg)), in a concentration-dependent manner, and voltage-dependent hysteresis of IK(erg) in GH3 cells. Moreover, SM-102, as TurboFectinTM and spermine, inhibited IK(erg) amplitude even when it was contained in the pipet. Similar results were obtained in MA-10 cells, where IK(erg) cells are involved in apoptosis and proliferation. Finally, they demonstrated that SM-102 reduces IK(IR) in microglial cells, but this result should be more extensive.
In my opinion, this article responds to the scope of the Biomedicines journal and should be of interest and useful for its readership. Anyway, the manuscript requires major revision, in order to be clearer and more exhaustive for the reader.
Major points
- In the results section, it is not reported how often do you applied the voltage-clamp protocol, such as voltage step protocol in figure 1.
- It is not clear when the different drugs are applied and consequently the voltage protocol. I mean, do you apply voltage step protocol for each 30sec? And when do you consider ctrl or drug effects? A single cell time course would be useful.
- In section 3.1.1. the authors said that both E-4031 and GLNVA inhibit the IK(erg) evoked by voltage-clamp protocol. Anyway, the authors have to show a figure (i.e. bars of the peak current and steady-state), about GLNVA also value in the text.
- In order to appreciate data dispersion, the mean values, such as peak amplitude, should represent as a scatter bar plot (it is not sufficient to report only the value ± SEM in the text).
- It is no clear the purpose to test SM-102 in microglia cells, please explain it and add some reference about this lipid and microglia, also in the introduction/discussion. In addition, SM-102 inhibits both inward and outward currents (not only KIR), please clarify.
- Since the authors have used only no-excitability cells, the phrase at the lines 467-472 is too speculative. Please better explain this and add a reference, otherwise remove it.
- As reported above, it would be appropriate to test SM-102 in apoptosis and/or proliferation assay.
- Figure 2B-C: there is any significance between ctrl and SM-102 for each membrane potential, such as at -60mV in figure 2C? the same in figure 5 B-C.
- Why do you show a representative instantaneous I-V relationship of IK(erg) in the presence of 100µM SM-102, in figure 3A? 300 µM SM-102 seems to be more efficacious.
Minor points
- Specify the statistical test used in figure 3B and 4B in the legends.
- It is useful to also show control (pipet without SM-102/TurboFectin/Spermine) at 1st and 20th pulses, in figure 4B.
- Please, add more references about the involvement of IK(erg) in apoptosis and proliferation in MA-10 cells.
- Please uniform the font size of the letters in all figures.
Author Response
Comments and Suggestions for Authors
Cho and colleagues present a paper entitled “Effective perturbations on the amplitude, gating and hysteresis of ergmediated potassium current caused by 1-octylnonyl 8-[(2- hydroxyethyl)[6-oxo-6-(undecyloxy)hexyl]amino]-octanoate (SM102), a cationic lipid”. * P 0 WORDS File Edit View Insert Format Tools Table Help Paragraph The authors have tested an ingredient present in the Moderna Covid-19 vaccine (SM-102), probably linked to the occurrence of myocarditis following the Covid-19 vaccination. In particular, they have evaluated SM-102 effects in GH (a cell line derived from a rat pituitary tumor), MA-10 (cell line derived from a mouse Leydig tumor) and microglia cells; all cells are not linked with myocarditis, anyway. However, the manuscript is well written and a detailed electrophysiological analysis was conducted. The authors demonstrated that SM-102 inhibits erg-mediated K current (I ), in a concentration-dependent manner, and voltage-dependent hysteresis of I in GH cells. Moreover, SM-102, as TurboFectin and spermine, inhibited I amplitude even when it was contained in the pipet. Similar results were obtained in MA-10 cells, where I cells are involved in apoptosis and proliferation. Finally, they demonstrated that SM-102 reduces I in microglial cells, but this result should be more extensive. In my opinion, this article responds to the scope of the Biomedicines journal and should be of interest and useful for its readership. Anyway, the manuscript requires major revision, in order to be clearer and more exhaustive for the reader.
Ans: Thanks for the comments provided by the reviewer.
Major points
- In the results section, it is not reported how often do you applied the voltage-clamp protocol, such as voltage step protocol in figure 1.
Ans: As commented by the reviewer, the rate of hyperpolarizing pulse used was set at 0.05 Hz. An additional Figure (Figure 1) depicts the time course for SM-102-mediated inhibition of IK(erg) detected in GH3 cells for the perusal.
- It is not clear when the different drugs are applied and consequently the voltage protocol. I mean, do you apply voltage step protocol for each 30sec? And when do you consider ctrl or drug effects? A single cell time course would be useful.
Ans: As advised by the reviewer, an additional Figure (i.e. time course of SM-102-induced inhibition of IK(erg)) was included in the revised manuscript (lines 241-242). The sequence of Figures was hence correspondingly changed.
- In section 3.1.1. the authors said that both E-4031 and GLNVA inhibit the I evoked by voltage-clamp protocol. Anyway, the authors have to show a figure (i.e. bars of the peak current and steady-state), about GLNVA also value in the text.
Ans: As commented by the reviewer, the data were hence included in the revised manuscript. That is, “For example, 1 min after GH3 were exposed to 10 mM E-4031, the peak or sustained IK(erg) was respectively decreased to 42 ± 6 or 21 ± 8 pA (n = 7) from control values of 177 ± 42 or 49 ± 9 pA (n = 7). Meanwhile, the presence of 10 mM GLNVA resulted in a reduction in the peak or sustained IK(erg) from 173 ± 39 or 47 ± 8 pA (n = 7) to 48 ± 7 or 23 ± 8 pA (n = 7), respectively.” (lines 223-227 in the revised manuscript).
- In order to appreciate data dispersion, the mean values, such as peak amplitude, should represent as a scatter bar plot (it is not sufficient to report only the value ± SEM in the text).
Ans: Thanks for the comments provided by the reviewer. We believed that the experimental results showing summary bar graphs (i.e., mean value ± SEM) would provide the sufficient information used for data interpretation, although the data dispersion might not be clearly represented.
- It is no clear the purpose to test SM-102 in microglia cells, please explain it and add some reference about this lipid and microglia, also in the introduction/discussion. In addition, SM102 inhibits both inward and outward currents (not only K ), please clarify
Ans: The main reason is that the intracellular dialysis with spermine can inhibit IK(IR) in microglial cells (lines 467-469 in the revised manuscript). Hence, we wanted to determine whether the presence of SM-102 can also perturb the magnitude of IK(IR) in these cells.
- Since the authors have used only no-excitability cells, the phrase at the lines 467-472 is too speculative. Please better explain this and add a reference, otherwise remove it.
Ans: As per the suggestion provided by the reviewer, the sentence “However, as this current was activated by long isosceles-triangular ramp pulse, the trajectory of IK(IR) found in BV2 microglial cells was not noticed to be hysteretical, suggesting that voltage-dependent hysteresis is not an intrinsic property in this current.” was removed from the Discussion section of the revised manuscript.
- As reported above, it would be appropriate to test SM-102 in apoptosis and/or proliferation assay.
Ans: Thanks for the comments provided by the reviewer. Such assay is important, and an additional sentence was hence included in the Discussion section of the revised manuscript. That is, “However, the detailed mechanism of SM-102 actions on IK(erg) as well as on apoptosis and/or proliferation assay still needs to be further delineated.” (lines 556-558).
- Figure 2B-C: there is any significance between ctrl and SM102 for each membrane potential, such as at -60mV in figure 2C? the same in figure 5 B-C.
Ans: The current amplitudes in Figure 2B between the absence and presence of 300 mM SM-102 were significantly different at the voltages more negative to -80 mV, while those in Figure 2C significantly differed at the voltages more negative to -50 mV (lines 308-311 in the revised manuscript). Moreover, current amplitudes in Figures 5B and 5C between the absence and presence of 300 mM SM-102 significantly differed at the voltages more negative to -60 mV (lines 448-450 in the revised manuscript).
- Why do you show a representative instantaneous I-V relationship of I in the presence of 100µM SM-102, in figure 3A? 300 µM SM-102 seems to be more efficacious.
Ans: Thanks for the reviewer’s comments. It needs to be mentioned that the voltage-dependent hysteresis of IK(erg) has been growingly noticed to play an important fact on electrical behaviors of electrically excitable cells. Hence, it would be useful to measure instantaneous current-voltage relationship of IK(erg) activated by the isoceles-triangular ramp pulse. Such non-equilibrium hysteretic behavior of IK(erg) is thought to rely on the previous state (conformation) of the Kerg channel due probably to dynamic voltage dependence or changes in the voltage gating of the channel. Alternatively, in the presence of 100 mM SM-102, the hysteretic area of IK(erg) demonstrated in this work was noticeably decreased. The hysteretic area obtained in the presence of 100 mM SM-102 was noted to be higher than that in 300 mM SM-102.
Minor points
- Specify the statistical test used in figure 3B and 4B in the legends.
Ans: As advised by the reviewer, the statistical methods were included in the legend of Figures 3B (lines 382-383) and 4B (lines 428-429).
- It is useful to also show control (pipet without SM-102/TurboFectin/Spermine) at 1 and 20 pulses, in figure 4B.
Ans: As per the suggestion provided by the reviewer, Figure 4B with respect to effect of intracellular dialysis with TurboFectinTM on IK(erg) was included (lines 407-409). The legend in Figure 4 of the revised manuscript was rephrased (lines 426-428).
- . Please, add more references about the involvement of IK(erg) in apoptosis and proliferation in MA-10 cells.
Ans: As advised by the reviewer, an additional reference was included in the revised manuscript (lines 436, reference #37).
- Please uniform the font size of the letters in all figures.
Ans: The font size in all Figures was appropriately modified.
Reviewer 2 Report
The current manuscript investigated the pharmacological effects of SM-102 on erg-17 mediated K+ current in several cell lines. The authors found that SM-102 can modulate the channel currents, I-V relationship, deactivation kinetics. Though the results are potentially interesting, there are still a few concerns which significantly weaken this manuscript.
Major concerns:
- It is still unclear how these different reagents including SM-102, TuboFectin, Spermine etc. Were they all dissolved in water? Was none of these dissolved in DMSO or other dissolvent? This message is quite important and should be unambiguously clarified.
- The title includes gating investigation. None of the figures presented in the manuscript demonstrated that the gating of the channel was modulated by SM-102. It is quite important to know whether SM-102 modulates the gating and such study should be carried out in all cell lines.
- Deactivation kinetics was only studied in figure 1 in cell line GH3. This kinetics should also be investigated in other cell lines as well to convince whether the SM-102 has a specific effect or a general effect to different cell lines.
- The dose-response curve for the inhibitory effect of SM-102 on the channel currents should also be provided in other cell lines than GH3 cell line.
- In figure 4, it would be inaccurate to use the same control currents for all other groups with different chemicals included in the recording pipette, as it is extremely unlikely that the control currents were actually exactly same for different groups.
- In figure 4 as well, the 10th and 20th recording curves were used to demonstrate the inhibitory effect of SM-102. There is no any experimental control to show that the current curves would stay stable without any chemicals included, which makes the accuracy of these results questionable. Also, the selection of 10th and 20th is very arbitrary without any justification. It seems that10th and 20th are just random choice. This needs to be justified with further experiments of time-dependent current curves after the application of the chemicals.
- There are no concrete evidence presented in this manuscript to support that all these recorded currents with the indicated protocols are strictly erg-17 mediated K+ current in all these cell lines. The accuracy results are therefore significantly weakened. Pharmacological approaches with multiple inhibitors need to be used at least to test this. Preferably the gene knock-down experiments should be carried out to convince this speculation. The experimental protocol used in current manuscript is very likely to open other type of channels including other types of potassium channels as well. This needs to be experimentally convinced in all cell lines.
- The results are mostly descriptive without any mechanism investigated. Further mechanistic insight into how such modulation or inhibition is achieved is needed to increase the importance and impact of this manuscript.
Author Response
Comments and Suggestions for Authors
The current manuscript investigated the pharmacological effects of SM-102 on erg-17 mediated K+ current in several cell lines. The authors found that SM-102 can modulate the channel currents, I-V relationship, deactivation kinetics. Though the results are potentially interesting, there are still a few concerns which significantly weaken this manuscript.
Ans: Thanks for the comments provided by the reviewer.
Major concerns:
- It is still unclear how these different reagents including SM-102, TuboFectin, Spermine etc. Were they all dissolved in water? Was none of these dissolved in DMSO or other dissolvent? This message is quite important and should be unambiguously clarified.
Ans: Thanks for the reviewer’s comments. The text was included in the revised manuscript (lines 110-114). That is, “SM-102 was dissolved in chloroform, spermine was in water, GLNVA was in dimethyl sulfoxide, while TurboFectinTM was in 80% ethanol solution.”
- The title includes gating investigation. None of the figures presented in the manuscript demonstrated that the gating of the channel was modulated by SM-102. It is quite important to know whether SM-102 modulates the gating and such study should be carried out in all cell lines.
Ans: Thanks for the comment provided by the reviewer. As noticed by the reviewer, the title in the revised manuscript was hence appropriately changed to “Effective perturbations of the amplitude and hysteresis of erg-mediated potassium current caused by 1-octylnonly-8-[(2-hydroxyethyl)[6-oxo-6 (undecyloxy)-hex-yl]amino]-octanoate (SM-102), a cationic lipid” (lines 2-5).
- Deactivation kinetics was only studied in figure 1 in cell line GH3. This kinetics should also be investigated in other cell lines as well to convince whether the SM-102 has a specific effect or a general effect to different cell lines.
Ans: As per the advice provided by the reviewer. An additional set of experiments was performed regarding the deactivation time course of IK(erg) in MA-10 Leydig cells. Hence, the experimental results were included in the text of the revised manuscript (lines 450-452). That is, “Moreover, during cell exposure to 300 mM SM-102, the tdeact value of IK(erg) activated from -10 to -110 mV was decreased to 106 ± 17 msec from a control value of 312 ± 22 msec. (n = 8, P < 0.05).”
- The dose-response curve for the inhibitory effect of SM-102 on the channel currents should also be provided in other cell lines than GH3 cell line.
Ans: As advised by the reviewer, the dose-response curve for the inhibitory effect of SM-102 on IK(erg) in MA-10 Leydig cells was carried out. The results were hence included in Figure 6D of the revised manuscript (lines 457-458). The text in the revised manuscript was hence rephrased.
- In figure 4, it would be inaccurate to use the same control currents for all other groups with different chemicals included in the recording pipette, as it is extremely unlikely that the control currents were actually exactly same for different groups.
Ans: Thanks for the comments pointed out by the reviewer. Hence, as advised by the reviewer, Figure 5C in the revised manuscript (lines 414-415) was hence redone, and the legend in Figure 5C was correspondingly rephrased.
- In figure 4 as well, the 10 and 20 recording curves were used to demonstrate the inhibitory effect of SM-102. There is no any experimental control to show that the current curves would stay stable without any chemicals included, which makes the accuracy of these results questionable. Also, the selection of 10 and 20 is very arbitrary without any justification. It seems that10 and 20 are just random choice. This needs to be justified with further experiments of time-dependent current curves after the application of the chemicals.
Ans: Thanks for the comments pointed out by the reviewer. An additional set of experiments regarding intracellular dialysis with CsCl. The text was hence included in the revised manuscript (lines 393-397). That is, “For the initial stage of measurements, we filled up the recording pipet with 140 mM CsCl, as membrane rupture occurred, the consecutive ramp pulse from -10 to -90 mV with a duration of 1 sec was applied at 0.01 Hz and, under our experimental conditions, the IK(erg) amplitude at 20th pulse was almost abolished.”
- There are no concrete evidence presented in this manuscript to support that all these recorded currents with the indicated protocols are strictly erg-17 mediated K+ current in all these cell lines. The accuracy results are therefore significantly weakened. Pharmacological approaches with multiple inhibitors need to be used at least to test this. Preferably the gene knock-down experiments should be carried out to convince this speculation. The experimental protocol used in current manuscript is very likely to open other type of channels including other types of potassium channels as well. This needs to be experimentally convinced in all cell lines.
Ans: Thanks for the reviewer’s comments. In attempts to reduce the concern raised by the reviewer, the concentration-dependent relationship of SM-102 effect on IK(erg) amplitude in MA-10 Leydig cells was performed. Alternatively, we did test different compounds (e.g., E-4031 and glyceryl noninvamide) for the inhibition of IK(erg) in GH3 and MA-10 cells. It would be interesting to perform gene-knock down experiments for testing if SM-102 could specifically block IK(erg). However, by using such maneuver, other types of potassium channels have the propensity to potentially be modified (either up- or down-regulated). Since the experimental conditions made in the present study are acute in onset and reversible; hence, other types of K+ currents which might influence the experimental results could be relatively negligible. An additional paragraph relevant to this issue was incorporated to the Discussion section of the revised manuscript (lines 548-558).
- The results are mostly descriptive without any mechanism investigated. Further mechanistic insight into how such modulation or inhibition is achieved is needed to increase the importance and impact of this manuscript.
Ans: Thanks for the comments pointed out by the reviewer. It needs to be emphasized that in our experimental conditions, we did provide for the first time the evidence that the voltage-dependent hysteresis of IK(erg) activated by the isosceles-triangular ramp pulse. Such non-equilibrium hysteresis of the current is essentially important in membrane excitability of varying cell types and it was noticed to be dynamically regulated by SM-102 or TurboFectinTM. Therefore, the experimental results that we have obtained in this study strongly suggest that the voltage sensor of Kerg channels could be potentially altered by SM-102. Therefore, any modifications of IK(erg) caused by the presence of SM-102 or TurboFectinTM are dependent not simply on the concentration of SM-102 or TurboFectinTM applied, but also on varying confounding factors that include the pre-existing resting potential or different firing patterns of action potentials existing in non-voltage-clamped cells occurring in vivo. Overall, findings from the present observations enable us to point out that the IK(erg) inherent in different cell types is an important target through which SM-102 or TurboFectinTM can act to influence the functional activities of the cells involved. However, the detailed mechanism of SM-102 actions on IK(erg) still needs to be further delineated.
Reviewer 3 Report
This is an interesting article, which should be accepted after a minor revision that should focus on the following points:
- The Author should provide structural formulas for the compounds they tested.
- The Authors should explain, why they decided to extend the studies on compounds other than SM-102 (Turbo-fectin, spermine), whereas only SM-102 is mentioned in the title.
- The Authors should provide more information about Kir channels in BV-2 microglial cells. Are these channels the same as Kir channels in heart cells ?
- The Authors mentioned that Kir channels in BV-2 cells are inhibited by internally applied spermine. However, no data is shown.
- The peak currents measured at -80 mV in Figure 3 are negative and they should be denoted in the text with a negative sign.
- As already mentioned, a moderate correction of English style is recommended.
Round 2
Reviewer 1 Report
Thanks for answering adequately to the requests. However, there are only minor points:
1. Please add the statistic test used in figure 3B-C and stars in the figure.
2. In figures 5 and 6, the letter sizes (A, B, C, D) are too large than the other figures.
3. The colors and the design of the figures are a bit confusing.
Author Response
Thanks for answering adequately to the requests. However, there are only minor points:
- Please add the statistic test used in figure 3B-C and stars in the figure.
Ans: Thanks for the comments raised by the reviewer. The statistical analysis was included in the text of Figure 3B-C and the asterisks were also added in Figure 3 (lines 325-330 in this revised version of the manuscript).
- In figures 5 and 6, the letter sizes (A, B, C, D) are too large than the other figures.
Ans: As advised by the reviewer, the letter sizes in Figures 5 and 6 were appropriately reduced.
- The colors and the design of the figures are a bit confusing.
Ans: The colors and design of Figures in the manuscript could be confusing to some extent. However, since the experimental results of ours appear to be sophisticated, we hope that the layout and design in the manuscript would be satisfactorily and intriguingly suitable for the readers, despite being incompletely unassailable.
Reviewer 2 Report
Most of my original concerns have already been addressed in the revised version, except this one: SM-102 dissolved in chloroform; spermine dissolved in water, GLNVA dissolved in dimethyl sulfoxide, TurboFectinTM dissolved in 80% ethanol solution.
Since different dissolvents were used for chemicals used in the study, it is important to know how the authors carried out the control experiments for each chemical, as some dissolvents may have side-effects out of the expectations.
Author Response
Most of my original concerns have already been addressed in the revised version, except this one: SM-102 dissolved in chloroform; spermine dissolved in water, GLNVA dissolved in dimethyl sulfoxide, TurboFectinTM dissolved in 80% ethanol solution.
Since different dissolvents were used for chemicals used in the study, it is important to know how the authors carried out the control experiments for each chemical, as some dissolvents may have side-effects out of the expectations.
Ans: Thanks for the comments provided by the reviewer. The text was appropriately changed. That is, “To make stock solution kept at -20 °C, SM-102 was dissolved in chloroform, spermine was in water, GLNVA was in dimethyl sulfoxide (DMSO), while TurboFectinTM was in 80% ethanol solution. During the experiments, chloroform, DMSO or ethanol in the desired concentration of each compound was diluted with bathing or pipette solution to be less than 0.01%.” (Lines 111-116 in the revised version of the manuscript).